# Oxidative Stress in Structural Valve Deterioration: A Longitudinal Clinical Study

**DOI:** 10.3390/biom12111606

**Published:** 2022-10-31

**Authors:** Manuel Galiñanes, Kelly Casós, Arnau Blasco-Lucas, Eduard Permanyer, Rafael Máñez, Thierry Le Tourneau, Jordi Barquinero, Simo Schwartz, Tomaso Bottio, Jean Christian Roussel, Imen Fellah-Hebia, Thomas Sénage, Arturo Evangelista, Luigi P. Badano, Alejandro Ruiz-Majoral, Cesare Galli, Vered Padler-Karavani, Jean-Paul Soulillou, Xavier Vidal, Emanuele Cozzi, Cristina Costa

**Affiliations:** 1Reparative Therapy of the Heart, Vall d’Hebron Research Institute (VHIR), Passeig Vall d’Hebron, 119, Autonomous University of Barcelona (UAB), 08035 Barcelona, Spain; 2Department of Cardiac Surgery, University Hospital Vall d’Hebron (HUVH)-ICS, Passeig Vall d’Hebron, 119, 08035 Barcelona, Spain; 3Infectious Diseases and Transplantation Division, Bellvitge Biomedical Research Institute (IDIBELL) and Bellvitge University Hospital-ICS, 08908 Barcelona, Spain; 4Department of Cardiac Surgery, Bellvitge University Hospital-ICS, 08907 Barcelona, Spain; 5Department of Cardiac Surgery, Quironsalud Teknon Heart Institute, 08022 Barcelona, Spain; 6CHU de Nantes, Institut du Thorax, Université de Nantes, CIC 1413, 44000 Nantes, France; 7CHU de Nantes, Institut du Thorax, Université de Nantes, CNRS, INSERM, 44000 Nantes, France; 8Gene and Cell Therapy, VHIR, UAB, 08035 Barcelona, Spain; 9Group of Drug Delivery and Targeting, Clinical Biochemistry Department, Vall d`Hebron Hospital Barcelona Campus, Vall d’Hebron Research Institute (VHIR), 08035 Barcelona, Spain; 10Department of Cardio Thorac Vascular Sciences and Public Health, Padua University Hospital, 35128 Padua, Italy; 11Department of Thoracic and Cardiovascular Surgery, Institut du Thorax, University Hospital, 44007 Nantes, France; 12Biostatistics Department, Université de Nantes, 44000 Nantes, France; 13Department of Cardiology, Vall d’Hebron Research Institut (VHIR), Hospital Vall d’Hebron, 08035 Barcelona, Spain; 14Department of Medicine and Surgery, University of Milano-Bicocca, 20126 Milan, Italy; 15Department of Cardiology, Neural and Metabolic Sciences, Istituto Auxologico Italiano, IRCCS, San Luca Hospital, 55100 Milan, Italy; 16Department of Cardiology, Bellvitge University Hospital, 08907 Barcelona, Spain; 17Avantea and Fondazione Avantea Onlus, 26100 Cremona, Italy; 18Department of Cell Research and Immunology, The Shmunis School of Biomedicine and Cancer Research, The George S. Wise Faculty of Life Sciences, Tel Aviv University, Tel Aviv 69978, Israel; 19Institut de Transplantation–Urologie–Néphrologie, INSERM Unité Mixte de Recherche 1064, Centre Hospitalier Universitaire de Nantes, 44093 Nantes, France; 20Department of Clinical Pharmacology, HUGH-ICS, 08001 Barcelona, Spain; 21Transplant Immunology Unit, Department of Transfusion Medicine, Padua University Hospital, 35128 Padua, Italy

**Keywords:** oxidative stress, aortic valve, biological heart valves, transcatheter aortic valve implantation, structural valve deterioration, protein nitration, lipid peroxidation

## Abstract

The cause of structural valve deterioration (SVD) is unclear. Therefore, we investigated oxidative stress markers in sera from patients with bioprosthetic heart valves (BHVs) and their association with SVD. Blood samples were taken from SVD (Phase A) and BHV patients during the first 24 (Phase B1) and >48 months (Phase B2) after BHV implantation to assess total antioxidant capacity (TAC), malondialdehyde (MDA), and nitrotyrosine (NT). The results show that MDA levels increased significantly 1 month after surgery in all groups but were higher at 6 months only in incipient SVD patients. NT levels increased gradually for the first 24 months after implantation in the BHV group. Patients with transcatheter aortic valve implantation (TAVI) showed even higher levels of stress markers. After >48 months, MDA and NT continued to increase in BHV patients with a further elevation after 60–72 months; however, these levels were significantly lower in the incipient and established SVD groups. In conclusion, oxidative stress may play a significant role in SVD, increasing early after BHV implantation, especially in TAVI cases, and also after 48 months’ follow-up, but decreasing when SVD develops. Oxidative stress potentially represents a target of therapeutic intervention and a biomarker of BHV dysfunction.

## 1. Introduction

Aortic valve replacement (AVR) is the standard therapy for significant aortic valve stenosis, representing the second most frequent cardiac operation after coronary artery bypass surgery. Furthermore, it is expected that the number of procedures will increase over the coming years because of the aging population and the rapid expansion of transcatheter aortic valve implantation (TAVI).

Two types of prosthetic valves are used for AVR, mechanical (MHV) and biological (BHV) heart valves, presenting both advantages and disadvantages [1]. While MHVs require lifelong patient anticoagulation with a higher risk of thromboembolisms or bleeding, BHVs have a better hemodynamic profile and do not require permanent anticoagulation. Therefore, compared with MHVs, BHVs are increasingly used. However, BHVs may undergo irreversible dysfunction known as structural valve deterioration (SVD). SVD increases over time and is more accelerated in young people who have a greater likelihood of reoperation, which carries a twofold increase in the risk of death [2]. Nevertheless, BHV may be a reasonable choice in patients above 50 years old [3]. Owing to the finite lifespan of BHVs and the rapid increase in people requiring aortic valve replacement, SVD remains the major concern regarding improving the clinical outcomes of these patients and reducing the significant burden on healthcare systems.

SVD is an intrinsic irreversible dysfunction of the prosthetic valve leaflets and/or supporting structures, but different definitions have been reported [4]. Recently, SVD has been classified into different stages (0 to 3) based on the degree of damage of the implanted prosthesis rather than on the patient’s clinical status [5]. For the present study, performed before the latest standardized definition, SVD was defined using the echocardiographic criteria previously used by our group [6].

SVD is usually the result of calcification of the valve cusps (intrinsic mineralization) and, less frequently, the result of degradation of the connective tissue matrix. However, the mechanisms of SVD remain unclear, although mechanical stress [7], inflammation [8], and an immune response to xenoantigens in BHVs [9,10] have been proposed to nonexclusively contribute as the initial triggers. The absence of anticalcification treatment [6,11,12] or the presence of prosthesis–patient mismatch [11] may lead to calcification of the leaflets and to SVD, and the wide use of glutaraldehyde as a fixative of the BHV leaflets does not prevent SVD. Furthermore, current valve processing does not eliminate valve immunogenicity, as suggested by the substantial amounts of galactose α1,3-galactose (αGal) [13] and *N*-glycolylneuraminic acid (Neu5Gc) [14] residues in BHVs even after glutaraldehyde fixation and by the swine major histocompatibility antigens (SLA-II) expressed in porcine valves after glutaraldehyde fixation [15]. The role of the immune response in SVD has been investigated and recently reported by us [10] in a European multicenter study (TransLink project).

Radical oxygen species (ROS) may also play a role in SVD, as suggested by the free radical species observed in native calcified aortic valves and BHVs [16,17,18]. Oxidative stress, caused by an imbalance between the production and degradation of ROS, is involved in many pathological processes, such as inflammation, ischemia/reperfusion, and infection, and may lead to the development of several pathological conditions [19]. Oxidative modifications of BHV leaflets, consistent with the formation of dityrosine collagen cross-links, oxidative modifications of amino acids, loss of glutaraldehyde cross-links, and collagen damage, have been identified in SVD [20,21]. ROS seem to play a central role as mediators of damage independent of the potential triggers for SVD (i.e., mechanical stress, inflammation/immune response) and, therefore, could be targets for diagnosis and prevention. Hence, the aim of this study was to investigate the presence of oxidative stress markers, namely, malondialdehyde (MDA), a reliable indicator of lipid peroxidation widely used in clinical studies, and nitrotyrosine (NT), an indirect marker of peroxynitrite, and the total antioxidant capacity (TAC) in patients with implanted BHVs with and without SVD who participated in the TransLink multicenter study (clinical trial number: NCT02023970).

## 2. Patients and Methods

### 2.1. Study Population

Patients diagnosed with SVD following implantation together with patients presenting with aortic valve disease in whom replacement with a BHV was indicated and those with an implanted BHV with >48 months of follow-up were invited to participate in the European collaborative project “Defining the role of xeno-directed and autoimmune events in patients receiving animal-derived bioprosthetic heart valves” (FP7/2007–2013; Grant Agreement 603049). In addition, patients undergoing aortic valve replacement with an MHV and those with coronary artery bypass graft (CABG) surgery were recruited to serve as controls (see protocol below). Patients who underwent these two operations with >48 months of follow-up also served as controls. The demographic data and the patients’ clinical characteristics were recorded.

The inclusion criteria were as follows: (i) age between 18 and 85 years at the time of surgery, (ii) AVR with a BHV or MHV with or without CABG surgery and TAVI with a BHV, and (iii) isolated CABG surgery. Patients were excluded from the study if they had previous cardiac surgery or presented severe renal failure (glomerular filtration rate (GFR) ≤ 30 mL/min/1.73 m^2^), severe dyslipidemia (total cholesterol > 350 mg/dL, triglycerides > 750 mg/dL), current cancer, or ongoing major infection.

### 2.2. Study Protocol

As shown in Figure 1a, the longitudinal study protocol featured two phases: (i) Phase A, or “diagnostic approach”, comprised patients exhibiting SVD (Cohort A), and (ii) Phase B, or “prospective approach”, included two cohorts, a Cohort B1, comprising de novo BHV recipients to characterize the kinetics of oxidative stress during the first 2 years after BHV implantation, and a Cohort B2, comprising a different set of patients with >48 months of BHV implantation to study the long-term oxidative stress response. This dual approach was required to cover the long period of time that in some cases may take SVD to develop so that the occurrence of the phenomenon could be studied within the time frame of 4.5 years that lasted the European project TransLink that funded this clinical trial. In both phases, patients with CABG surgery performed under cardiopulmonary bypass and MHV replacement served as controls.

### 2.3. Procurement of Samples and Informed Consent

Blood samples were taken at different time points: Phase A—at the time of SVD diagnosis; Phase B1—before surgery and 1, 6, 12, and 24 months after surgery; and Phase B2—at the time of recruitment (>48 months after surgery) and 12–24 months later (>60–72 months after surgery). In all these cases, serum samples were obtained and stored at −80 °C until used for analyses. The patients in the CABG and MHV groups in Phase B1 were sampled at the same time points as those in the BHV group, and in Phase B2, a single blood sample was taken at the time of recruitment >48 months after surgery. Figure 1b shows the number of participants in the study from whom informed consent was obtained (PR314/2013). The research was performed in accordance with the principles outlined in the Declaration of Helsinki.

### 2.4. Echocardiographic Assessment and Definition of Structural Valve Disease (SVD)

The patients were monitored with echocardiography at different time points throughout the studies, and the interpretations were carried out and analyzed by experienced investigators (T.L.T., A.R., A.E., and L.B.) on commercial ultrasound systems (GE Vivid series, Waukesha, WI, USA; or Philipps, Andover, MA, USA) and stored (ImageVault and EchoPAC software, GE Medical Systems, Horten, Norway) in a centralized CoreLab (Nantes, France). As previously defined by us [6], SVD was diagnosed when one or more of the following features were present in the echocardiogram: (i) mean aortic prosthesis (MAP) gradient ≥ 30 mmHg, (ii) prosthesis effective orifice area ≤ 1 cm^2^ that worsened over time, or (iii) intraprosthesis aortic regurgitation ≥ 2/4. Endocarditis, perivalvular leakage, and pannus overgrowth were not considered SVD. During follow-up, early or incipient SVD was defined according to current recommendations [4,5] as an incremental (Δ) MAP gradient ≥ 10 mmHg. Patients in Group B2 who developed SVD during follow-up were kept in Group B2.

### 2.5. Assessment of Total Antioxidant Capacity (TAC)

The measurement of TAC was used because it is more informative than measuring the levels of specific natural antioxidants. The TAC was determined using the OxiSelect™ TAC Assay Kit (Cell Biolabs, San Diego, CA, USA) according to the manufacturer’s instructions. The assay is based on the reduction of copper from Cu^2+^ to Cu^+^. Briefly, samples and standards were diluted with a reaction reagent, and upon the addition of copper, the reaction proceeded for just a few minutes. The TAC was determined by comparing the reduction of copper to the reducing activity of uric acid standards, and the results were then expressed as mM.

### 2.6. Assessment of Malondialdehyde (MDA)

The MDA level was determined by adding 25 µL BHT 0.5%, 2.5 µL EDTA 0.14 mM, 50 µL TCA 40%, and 200 µL TBA 0.67% to 50 µL of sample and heating at 100 °C for 10 min. Then, glacial acid and chloroform were added to the sample, and the light emission was measured with a photometer (Microplate Photometer, Thermo Fisher Scientific Multiskan FC, Waltham, MA, USA) at 530 nm. The results are expressed as µM.

### 2.7. Assessment of Nitrotyrosine (NT)

The NT level was determined using the OxiSelect™ Nitrotyrosine ELISA Kit (Cell Biolabs, San Diego, CA, USA), according to the manufacturer’s instructions, and the results are expressed as nM.

### 2.8. Statistical Analyses

Continuous data are reported as the mean and standard deviation or the median and interquartile range according to the data distribution, and categorical data are reported as frequencies and percentages. To account for the skewed data distribution, the results are graphically presented as boxplots. For the same rationale, as the values of the main variables (TAC, MDA, and NT) fit a lognormal distribution well, generalized linear mixed models were applied for the analyses, where the patient was considered a random effect. An exponential temporal covariance structure was used to account for the uneven time points when needed. Multiple comparison adjustments were performed with Dunnett’s and Tukey–Kramer’s procedures. A two-sided *p* < 0.05 was considered statistically significant. All statistical analyses were performed using SAS^®^ 9.4 (SAS Institute Inc., Cary, NC, USA).

## 3. Results

Figure 1b displays the number of patients sampled in the different phases and study groups, Table 1 summarizes the characteristics of all the patients enrolled in the study, and Table 2 and Appendix A describe the types of implanted BHVs and MHVs, respectively. They show that the BHVs were implanted surgically (AVR) in the majority of the patients, but some patients were subjected to TAVI.

### 3.1. Oxidative Stress during the First 24 Months after BHV Implantation (Phase B1)

As shown in Figure 2a, the TAC values before surgery (baseline) were similar in all three groups. After surgery, differences between the values at each sampling time compared with their own baseline values were not statistically significant when compared between groups.

By contrast, Figure 2b shows that the MDA values before surgery tended to be lower in the MHV group. Nevertheless, the MDA values were significantly greater in all study groups 1 month after surgery when compared with the corresponding baseline values (*p* < 0.001, *p* < 0.001, and *p* = 0.009 for BHVs, CABGs, and MHVs, respectively). Thereafter, the values in all groups were not significantly different from baseline, suggesting that the observed 1-month increase may be attributed to a response to surgery. When the groups were compared at the various later sampling times, there was a significant difference between the BHV and MHV groups at 12 months only (*p* = 0.020).

Figure 2c shows that the NT values increased following surgery by 24 months after surgery, although the values were similarly elevated in all the study groups (*p* < 0.001 versus their own baseline values in all instances).

Notably, there were no significant differences in the TAC, MDA, and NT values between the patients with different BHVs (bovine, equine, and porcine; data not shown). Interestingly, the comparison displayed in Figure 3a–c of AVR patients with pericardium BHVs versus the TAVI group (with BHVs made of pericardial tissue) shows that the MDA levels increased significantly 1 month after surgery (*p* < 0.001 versus baseline) in the AVR group but not in the TAVI group; however, they were significantly higher in the TAVI than in the AVR group by 12 months after implantation (*p* < 0.001). Furthermore, the NT levels started to increase earlier in TAVI patients (after 6 months, *p* = 0.0144 versus baseline and *p* = 0.003 versus AVR group) with further elevation after 12 months of the follow-up (*p* = 0.002 versus baseline and *p* = 0.01 versus AVR group; see Figure 3c). Of note is that although the NT levels in the AVR group were also significantly increased after 12 and 24 months (*p* < 0.001 versus baseline in both cases), values were always higher in the TAVI group.

### 3.2. Oxidative Stress in Incipient SVD (Phase B1)

Patients with a BHV and exhibiting an increase in the MAP gradient ≥ 10 mmHg (e.g., incipient SVD) were already identified in Phase B1. In particular, 14 patients were identified with a Δ MAP gradient ≥ 10 mmHg between the 6- and 12- to 24-month follow-ups, and 9 patients had serum samples available for the assessment of oxidative stress status. All were AVR patients and none TAVI. These were compared with the 92 patients showing a MAP gradient <10 mmHg (e.g., without incipient SVD) in which oxidative stress status was also determined at various time points (baseline and 1, 6, and 12–24 months). Therefore, in the analysis, each patient acted as his or her own control. Figure 4a shows that the TAC values tended to increase by 6- and 12- to 24-month follow-ups in both B1 groups, reaching statistical significance in the group without incipient SVD at 12–24 months (*p* = 0.011 vs. baseline and *p* = 0.005 vs. 1-month follow-up). The MDA results displayed in Figure 4b show that by the 6-month follow-up, values were significantly greater (*p* = 0.018) in the incipient SVD group than in the group without incipient SVD. Thereafter, by the 12–24-month follow-up, the values were similar in both groups. Figure 4c shows that the NT values were not significantly changed during the first 6 months after AVR; however, by 12–24 months of follow-up, they increased in the two study groups.

### 3.3. Oxidative Stress Associated with SVD (Phase A) and BHVs >48 Months after AVR (Phase B2)

To compare the degree of oxidative stress in patients with SVD and in patients with a BHV implanted >48 months prior without evidence of SVD, the serum levels of TAC, MDA, and NT were determined at inclusion. Again, patients who underwent an MHV implantation or CABG surgery >48 months prior served as controls.

Figure 5a demonstrates that TAC values in patients with SVD were similar to the values found in the BHV and CABG groups; although, they seemed lower in the MHV group than in all the other groups.

The MDA results shown in Figure 5b demonstrate lower values in the SVD group than in the BHV group (*p* = 0.025). They also tended to be lower in the SVD group than in the CABG and MHV groups, but the differences were not significant. Finally, the NT results were not associated with significant differences between groups (Figure 5c).

### 3.4. Oxidative Stress Changes in the Follow-Up of Phase B2 Patients

To assess whether patients with a BHV implanted >48 months prior (median: 6.6 years; IQR: 5.6–8.0) undergo an increase in oxidative stress that could be associated with the development of SVD, the sera from those who had a second follow-up visit 12–24 months after (>60–72 months after BHV implantation) were used to determine the TAC, MDA, and NT values, and the results were compared with those at the time of recruitment (>48 months after BHV implantation). The results shown in Figure 6a–c demonstrate that after 12–24 months of follow-up, the TAC, MDA, and NT values were significantly increased (*p* < 0.001, *p* = 0.02, and *p* < 0.001, respectively). When a paired comparison was made, taking only the results of the patients who were sampled at the two times, an identical picture emerged between the B2 baseline values (>48 months after BHV implantation) and the 12- to 24-month values (>60–72 months after BHV implantation) (data not shown). These results clearly demonstrate that beyond 48 months after BHV implantation, patients exhibit a significant increase in oxidative stress markers.

### 3.5. Effect of the Source of BHV Tissue on Oxidative Stress (Phase B2)

The results for patients in whom bovine pericardium and porcine aortic valves were implanted >48 months prior were compared at baseline (>48 months after BHV implantation) and 12–24 months later (>60–72 months after BHV implantation). As shown in Figure 6d, the TAC values were similar in the patients with implanted bovine pericardium and porcine aortic valves when first assessed >48 months after BHV implantation. However, after 12–24 months of follow-up, the levels increased significantly in the bovine pericardium group (*p* < 0.001), while they did not change significantly in the porcine aortic valve group.

Figure 6e shows that the MDA values were again similar in the two valve tissue groups >48 months after surgery; however, after 12–24 months of follow-up (>60–72 months after BHV implantation), the values increased significantly in the bovine pericardium group (*p* = 0.007) and tended to decrease in the porcine aortic group, although the difference was not significant.

The results for NT shown in Figure 6f also demonstrate that the values in the bovine pericardium and porcine aortic valve groups after >48 months of prosthesis implantation were not significantly different; however, after 12–24 months of follow-up (>60–72 months after BHV implantation), the values had increased significantly in both BHV groups (*p* < 0.001 and *p* = 0.015, respectively, in comparison with their respective baseline values).

### 3.6. Oxidative Stress in Incipient SVD (Phase B2)

Of the 477 patients in Phase B2 in whom oxidative stress was assessed at inclusion, a second assessment after 12–24 months of follow-up (>60–72 months after BHV implantation) was performed in 79 patients, and of these patients, 11 were identified with a Δ MAP gradient ≥10 mmHg between baseline and follow-up. The results of oxidative stress markers from these two sets of patients were compared at both time points and with the results for those diagnosed with established SVD (Phase A). Figure 7a shows that the TAC values were similar in both B2 groups at baseline and then increased after 12–24 months of follow-up; a significant difference (*p* = 0.001) was observed only in the group without incipient SVD. Of note was the lower TAC values in the patients with established SVD compared with the B2 group without incipient SVD at the 12- to 24-month follow-up (*p* = 0.024). The MDA results shown in Figure 7b demonstrate a lower baseline value in the group that developed incipient SVD compared with the group without incipient SVD (*p* = 0.009), with a similar trend at the 12- to 24-month follow-up, although at this time point, the difference was not significant. Again, lower values were observed in the group with established than in the group without incipient SVD at baseline and at the 12- to 24-month follow-up (*p* = 0.01 and *p* < 0.001, respectively) but were similar to the group with incipient SVD. The NT results shown in Figure 7c demonstrate a significant increase from baseline to 12–24 months of follow-up in the group without incipient SVD (*p* < 0.001); these values were also significantly greater than those observed in the established SVD group (*p* < 0.001). The NT values in the group with incipient SVD were similar to those in the established SVD group and tended to be lower than those in the group without incipient SVD, although the difference was not significant.

### 3.7. Effect of BHV Framework Type and Size on Oxidative Stress (Phase B2)

The patients with bovine pericardium valves implanted >48 months prior were separated into three groups according to their framework: stented, stentless, and TAVI. As all porcine aortic valve prostheses are stented, this analysis was not performed for this type of BHV. The results obtained at the time of recruitment (>48 months after BHV implantation) were compared between them and with the values measured 12–24 months later (>60–72 months after BHV implantation). Figure 8a shows that the TAC values were not significantly different between the three groups at >48 months after BHV implantation. However, after 12–24 months of follow-up, the TAC values increased to a similar degree in the subjects with stented and stentless BHVs, with a significant difference observed for the former (*p* < 0.001) but not the latter. The MDA and NT results shown in Figure 8b,c, respectively, also demonstrate that after the 12- to 24-month follow-up, the two oxidative stress markers exhibited significant increases in the stented group (*p* = 0.011 and *p* < 0.001, respectively). The number of patients with stentless BHVs and TAVI was too small to perform statistical comparisons.

Appendix A shows a greater incidence of SVD in individuals with Mitroflow BHVs, especially in those with small diameters (19–21 mm). However, as shown in Appendix A and Appendix A, there were no significant differences in oxidative stress markers between 19–21 mm and >21 mm diameter at any time point in Phases B1 and B2.

### 3.8. Effect of Comorbidities and Medical Treatments on Oxidative Stress

The study of the effect of comorbidities on oxidative stress in patients with >48 months of BHV implantation (Phase B2) shown in Appendix A demonstrated a significant difference for NT in patients with ischemic heart disease. However, no significant differences were found between groups for the presence of comorbidities, such as diabetes mellitus, arterial hypertension, and obesity.

Appendix A show that the treatment with the antiglycemic drug metformin had a significant effect on the NT but not in the MDA serum levels. NT values were significantly lower in patients treated with metformin than in those without treatment before BHV implantation and during the first 6 months after implantation. However, beyond the 12-month follow-up, the NT increased to a similar degree in both groups, a trend that was maintained for the remainder of the follow-up (Phase B2 patients). Interestingly, once SVD developed (Phase A), the NT was reduced again to values similar to those seen at baseline with a greater reduction in the metformin-treated group, although the difference did not achieve statistical significance (*p* = 0.079). Other medications did not show a noticeable effect on oxidative stress markers (data not shown).

## 4. Discussion

This longitudinal multicenter clinical study is the first to demonstrate that oxidative stress markers increase with time in the serum of patients with a BHV and decrease to baseline levels when incipient and established SVD develop. These findings are of clinical relevance and support the hypothesis that ROS play a key role as early effectors of SVD and warrant further investigation and discussion.

### 4.1. Oxidative Stress Increases after BHV Implantation and Decreases Once SVD Develops

Our results show that an increase in oxidative stress markers is noticeable in serum 6 months after BHV implantation and that a further increase occurs beyond the 48 months after implantation, as illustrated by the significant increases in MDA and NT levels and the concomitant increase in TAC values. The elevation in TAC values is likely associated with a compensatory mechanism to counteract oxidative stress [22,23]. By contrast, a marked reduction in oxidative stress markers was observed in Phase B2 incipient (Δ MAP gradient ≥ 10 mmHg) and established (MAP gradient ≥ 30 mmHg) SVD to levels similar to those found prior to BHV implantation. These findings support the hypothesis that ROS play a role as mediators of SVD, which is also supported by the oxidative modifications found in calcific aortic sclerosis [24,25] and in the leaflets of BHVs with SVD [19,20]. In addition, dysregulation of glutathione homeostasis has been associated with aortic valve sclerosis [26]. Indeed, in vitro studies have shown that oxidative stress promotes calcification [27] via enhanced BMP2 expression and signaling [28].

It is worth pointing out that the transient increase in lipid peroxidation observed 1 month after cardiac surgery in our study, occurring both in the patients with BHVs and in the controls, is likely caused by the anesthesia and surgery, as the TAVI group not subjected to surgery did not exhibit such elevation. This hypothesis is supported by studies from our own laboratory [29,30] and by other investigators [31] showing increased oxidative stress during cardiac surgery and during the early postoperative period and its association with a greater incidence of postoperative atrial fibrillation [31].

The lower levels of oxidative stress markers observed in the patients with incipient and established SVD (>48 months) in comparison with the patients without SVD suggest that the cause of the oxidative stress is within the leaflets of the BHV and that once they become calcified, the oxidative stress trigger wears off. This thesis is further supported by the finding that the antiglycemic drug metformin reduces NT levels before BHV implantation and during the first 6 months of follow-up. Thereafter, the NT values continue to rise with the disappearance of the beneficial effect of metformin, only to be reduced again when SVD develops. Indeed, the antioxidant effect of metformin has been previously described in endothelial cells [32] and in patients with type 2 diabetes [33].

The most plausible cause of the observed oxidative stress changes is an immune response to the presence of xenoantigens on the BHV implants. The immunogenicity hypothesis for SVD is supported by the reported substantial amounts of αGal residues remaining in BHVs [13], even after combined aggressive decellularization protocols [34], and by the abundant expression of swine major histocompatibility antigens (SLA-II) in BHVs despite decellularization and glutaraldehyde fixation [15]. Recently, we also have shown that Neu5Gc is expressed in several commercial BHVs and that they are recognized by human anti-Neu5Gc IgG [14]. Notably, recognition of bound antibodies through Fcγ receptors is a key trigger of neutrophil and macrophage activation, phagocytic activity, and ROS production [35,36]. Animal models using αGal-deficient mice [37] and Neu5Gc-deficient mice [10], generated by knockout of the genes GGTA1 and CMAH that encode for the enzymes that catalyze their synthesis, also suggest that the immune response against αGal- and Neu5Gc-positive valves can induce calcification and deterioration of experimental implants. Hence, as xenoantigens are not fully removed from BHVs, they may also affect the valve implant. Notwithstanding, if the immune response triggers oxidative stress, then the recent procurement of knockout cattle and pigs lacking αGal and Neu5Gc [38,39] may result in a reduction in the oxidative stress in clinical settings through a reduction in the immunogenicity burden in BHVs.

### 4.2. Oxidative Stress Is Greater in TAVI Patients

A greater increase in oxidative stress markers was observed in Phase B1 in TAVI patients as compared with AVR patients, in whom the BHV was also made of pericardial tissue. A possible explanation for these results may be the known damage that TAVI valves undergo by “crimping” during the implantation procedure [40], a damage that may be responsible for an increase in oxidative stress and that, in turn, may lead to calcification of the prosthesis [41]. In this connection, some investigators have reported a gradual increase in calcification in TAVI valves, progressing to valve degeneration within the first 5 years of implantation [42], also making them more prone to valve thrombosis [43]. The number of patients with TAVI was not sufficient to demonstrate these clinical outcomes, but this aim was out of the scope of the present study.

### 4.3. Influence of BHV Tissue Type and Size on Oxidative Stress

The present findings show that there is a distinctive response between bovine pericardium and porcine aortic BHVs, with NT levels increasing in patients carrying both BHV types and MDA levels increasing only in those with bovine aortic valves. A potential explanation for the differential oxidative stress response between the two BHV types is the greater collagen content observed in the implants made of pericardial tissue compared with porcine aortic tissue [44], as platelets become activated upon exposure to collagen and have a potential proinflammatory action [45]. We have previously shown, using cultured human leukocytes, that protein nitration is predominantly generated by a peroxynitrite (ONOO)-mediated pathway [46] and that ONOO^−^ plays an essential role in the mechanism of proinflammatory cytokine release by monocytes, an effect that is necessarily mediated by the activation of the Rel/NF-kB pathway [47]. Indeed, inflammation has been found to be associated with the calcification of aortic valves [8,48]. Furthermore, interleukin (IL)-6, a soluble mediator with pleiotropic effects, is increased in human calcific aortic valves, where it promotes mineralization of valve interstitial cells via the expression of BMP2 [49]. Similarly, the proinflammatory cytokine IL-1β has been reported to increase in stenotic aortic valve leaflets [50]. Therefore, cytokines might be key inflammatory mediators of the osteogenic program in aortic valves through multiple pathways, and they may link immunogenicity and oxidative stress mechanisms in SVD. Taken together, the xenoantigens present in BHVs together with other proinflammatory factors may induce the production of xenoantibodies and an inflammatory response that, in turn, leads to an increase in oxidative stress. Once the BHV calcifies, the xenoantigens are removed from the tissues, causing the dissipation of the immune and inflammatory response and the reduction in oxidative stress.

Notwithstanding, the best tissue type for BHVs remains unclear as clinical studies have reported variable BHV durability depending on the animal origin of the tissue, with some studies showing a better outcome with bovine pericardium than with porcine pericardium [51], others demonstrating similar durability between them [52], and even a few reporting good long-term durability of porcine aortic valve prostheses [53]. It is clear that this is an issue that requires further investigation.

Since stentless BHVs have better hemodynamic performance than stented BHVs [54], the lack of difference in the degree of oxidative stress associated with these two types of BHVs in our study supports the hypothesis that oxidative stress is independent of flow changes, a thesis that is further supported by the absence of differences in oxidative stress markers between small (19–21 mm diameter) and bigger-size (>21 mm diameter) BHVs.

### 4.4. Study Limitations

One limitation of the study was related to the BHV type used, which varied depending on the participating centers and on the surgeon’s preference. As a result, not all BHV types were investigated, and the number of BHVs in each group varied between the types. However, it should be recognized that this kind of study is difficult to perform, and therefore, the protocol design was adapted to the particular clinical features of AVR. Another limitation was the assessment of oxidative stress markers in available serum samples after completion of the primary investigation, consisting of analyzing the antibody response in BHVs [10]. Because of this, serum samples were not obtained from all the patients. Furthermore, despite the strong statistical differences observed, an additional shortcoming inherent to clinical studies such this is the difficulty of demonstrating a cause–effect link. Therefore, although the results obtained in the present study strongly support the hypothesis that oxidative stress plays a significant role in SVD, further studies are required to confirm this relationship.

## 5. Conclusions

The identification of increased oxidative stress markers in BHV patients as early as 6 months after implantation, with further elevation beyond 48 months after implantation, but decreased oxidative stress when SVD develops supports the hypothesis that oxidative stress in the BHV plays a significant role in SVD. These findings have important clinical implications since they open the possibility that antioxidants can become potential therapeutic agents for the prevention or delay of SVD. Furthermore, this study opens the door for the potential use of oxidative stress biomarkers for the early detection of SVD.

## Figures and Tables

**Figure 1 biomolecules-12-01606-f001:**
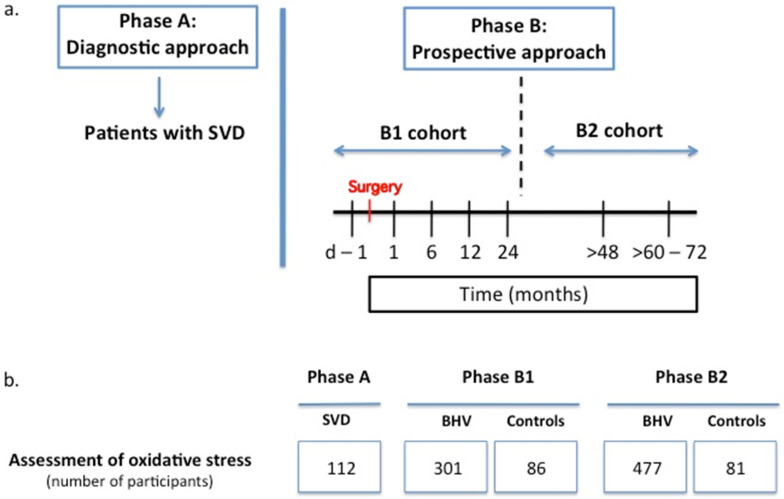
(**a**) Study protocol diagram (SVD—structural valve deterioration). See text for details. (**b**) Chart showing the number of patients recruited for the assessment of oxidative stress in the different phases and study groups (BHV, bioprosthetic heart valve).

**Figure 2 biomolecules-12-01606-f002:**
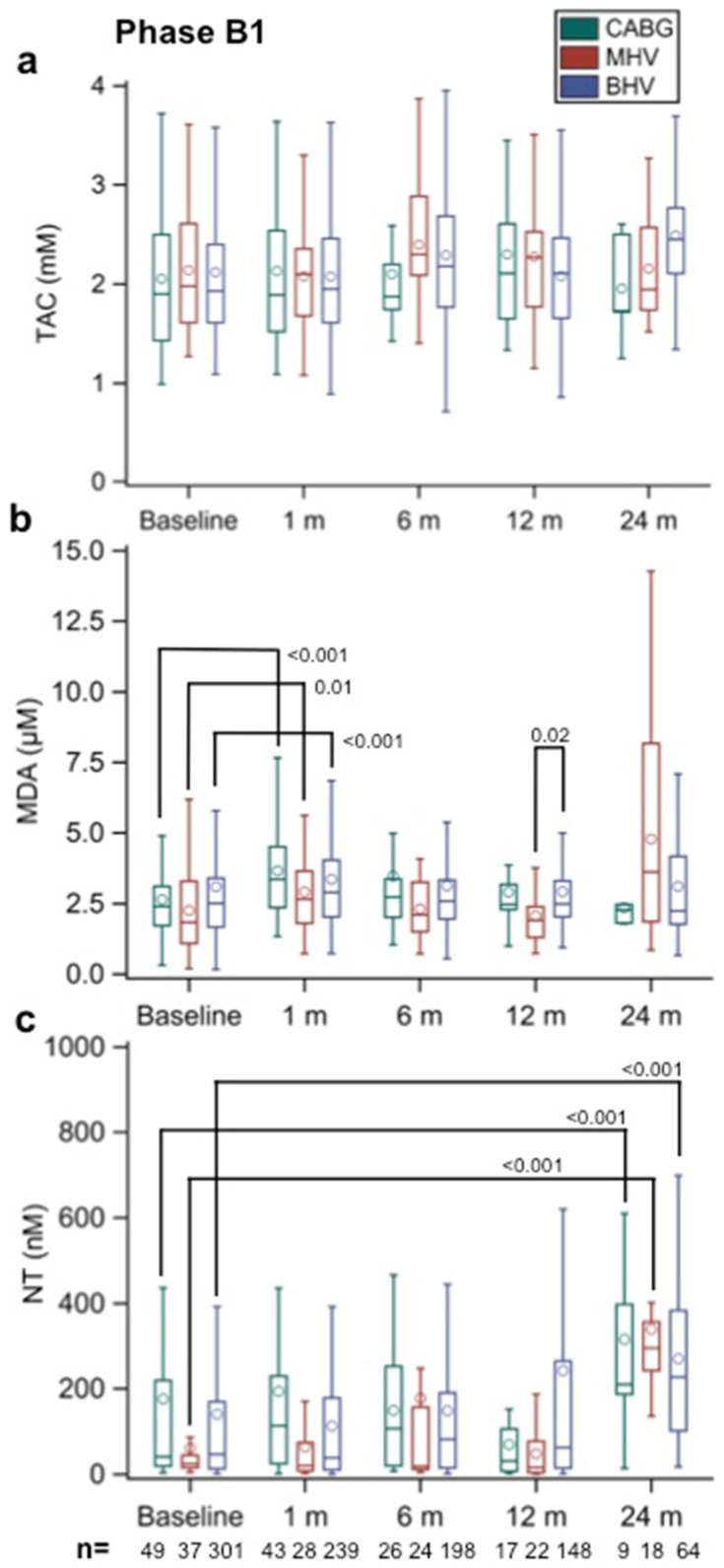
(**a**) Total antioxidant capacity (TAC, in mm), (**b**) malondialdehyde (MDA, in µm), and (**c**) nitrotyrosine (NT, in nm) values in Phase B1 patients with implanted aortic biological heart valves (BHVs) at different time points (baseline and 1, 6, 12, and 24 months after implantation). Patients undergoing coronary artery bypass (CABG) surgery and aortic valve replacement with a mechanical heart valve (MHV) served as the controls. The numbers within the figures represent the *p*-values, and the numbers at the bottom of the figures represent the number of assessed patients. Multiple comparison adjustments were performed with Dunnett’s and Tukey–Kramer’s procedures.

**Figure 3 biomolecules-12-01606-f003:**
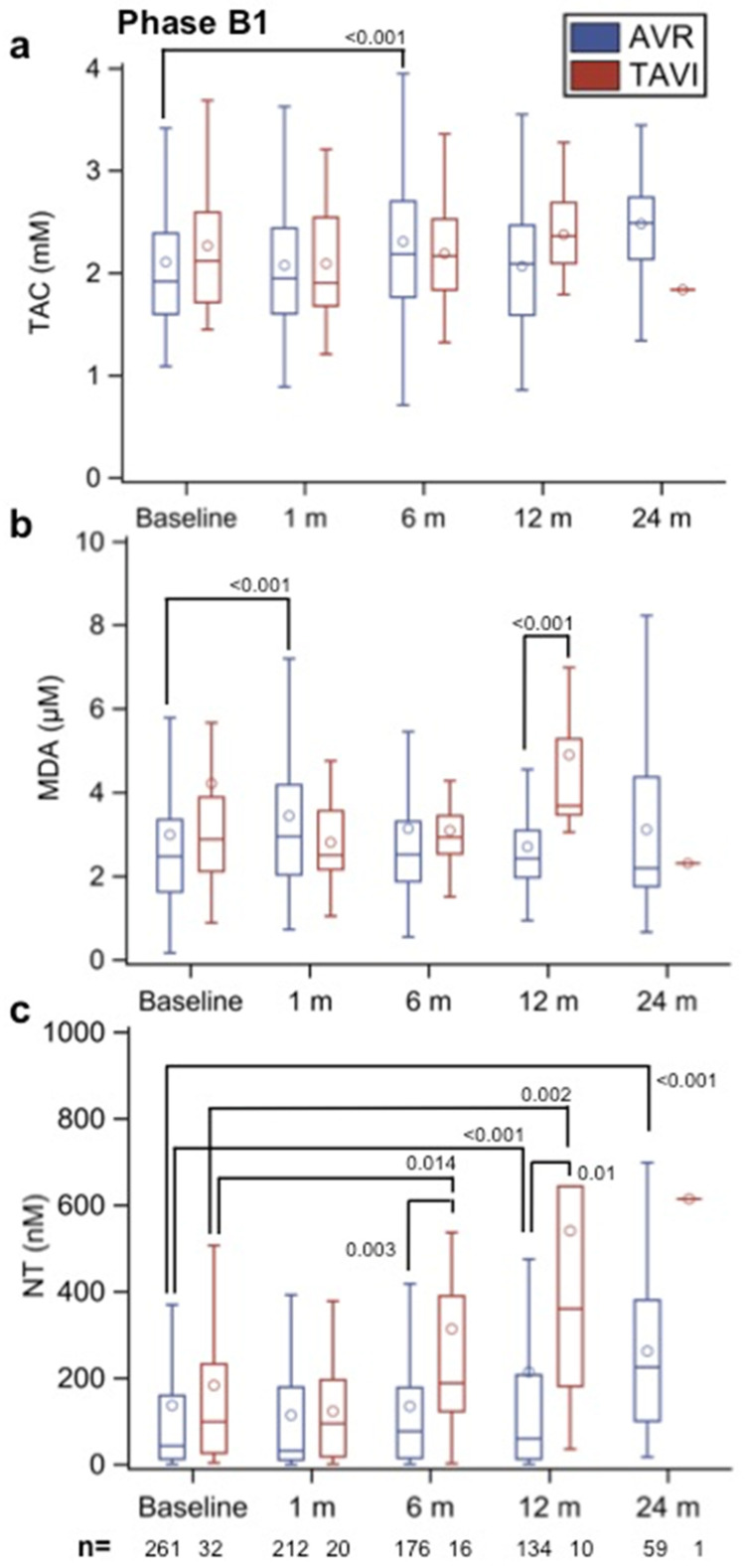
(**a**) Total antioxidant capacity (TAC, in mm), (**b**) malondialdehyde (MDA, in µm), and (**c**) nitrotyrosine (NT, in nm) values in Phase B1 patients with aortic biological heart valves (BHVs) only made of pericardial tissues implanted surgically (AVR) and by transcatheter aortic valve implantation (TAVI) at different time points (baseline and 1, 6, 12, and 24 months after implantation). The numbers within the figures represent the *p*-values, and the numbers at the bottom of the figures represent the number of assessed patients. Multiple comparison adjustments were performed with Dunnett’s and Tukey–Kramer’s procedures.

**Figure 4 biomolecules-12-01606-f004:**
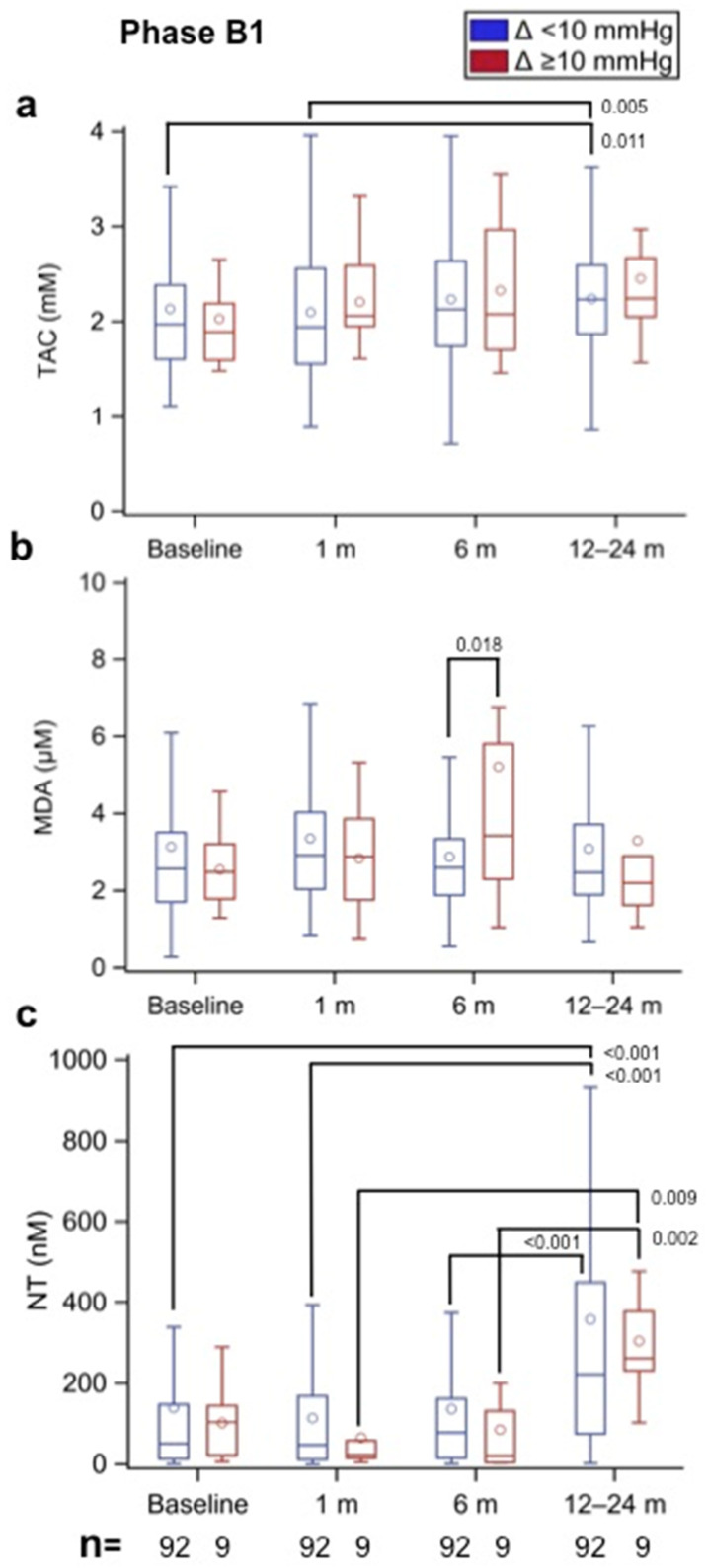
(**a**) Total antioxidant capacity (TAC, in mm), (**b**) malondialdehyde (MDA, in µm), and (**c**) nitrotyrosine (NT, in nm) values in Phase B1 patients with implanted aortic biological prostheses at different time points according to the observed increment (Δ) in the mean aortic prosthesis gradient (Δ < 10 mmHg—without incipient SVD, and Δ ≥ 10 mmHg—with incipient SVD) between the 6- and the 12- to 24-month follow-ups. Therefore, each patient acted as his or her own control (e.g., paired results). The numbers within the figures represent the *p*-values, and the numbers at the bottom of the figures represent the number of assessed patients. Multiple comparison adjustments were performed with Dunnett’s and Tukey–Kramer’s procedures.

**Figure 5 biomolecules-12-01606-f005:**
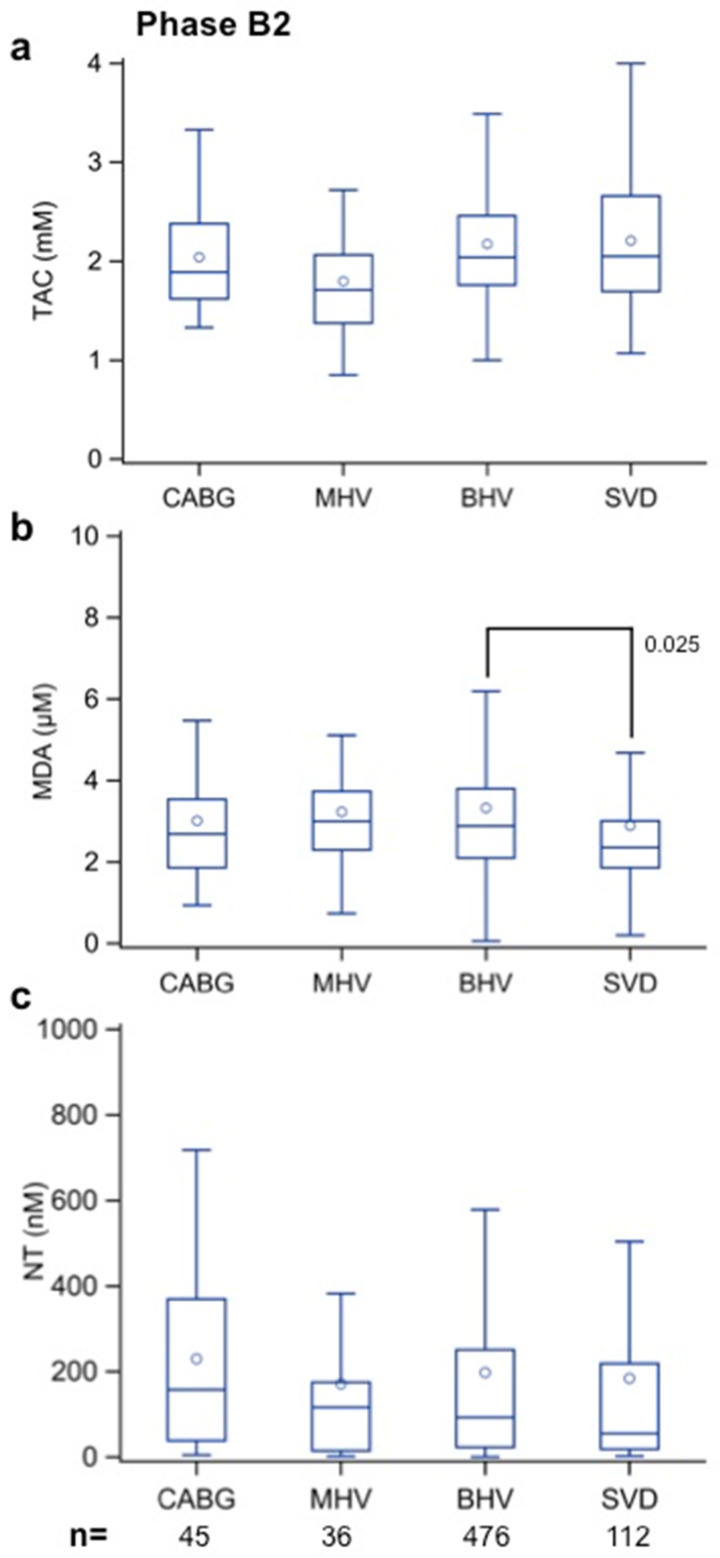
(**a**) Total antioxidant capacity (TAC, in mm), (**b**) malondialdehyde (MDA, in µm), and (**c**) nitrotyrosine (NT, in nm) values in patients diagnosed with structural valve disease (SVD—Phase A) and in Phase B2 patients with implanted aortic biological heart valves (BHV) at the time of recruitment (>48 months after surgery). Patients undergoing coronary artery bypass (CABG) surgery and aortic valve replacement with a mechanical heart valve (MHV), also recruited >48 months after surgery, served as the controls. The numbers within the figures represent the *p*-values, and the numbers at the bottom of the figures represent the number of assessed patients. Multiple comparison adjustments were performed with Dunnett’s and Tukey–Kramer’s procedures.

**Figure 6 biomolecules-12-01606-f006:**
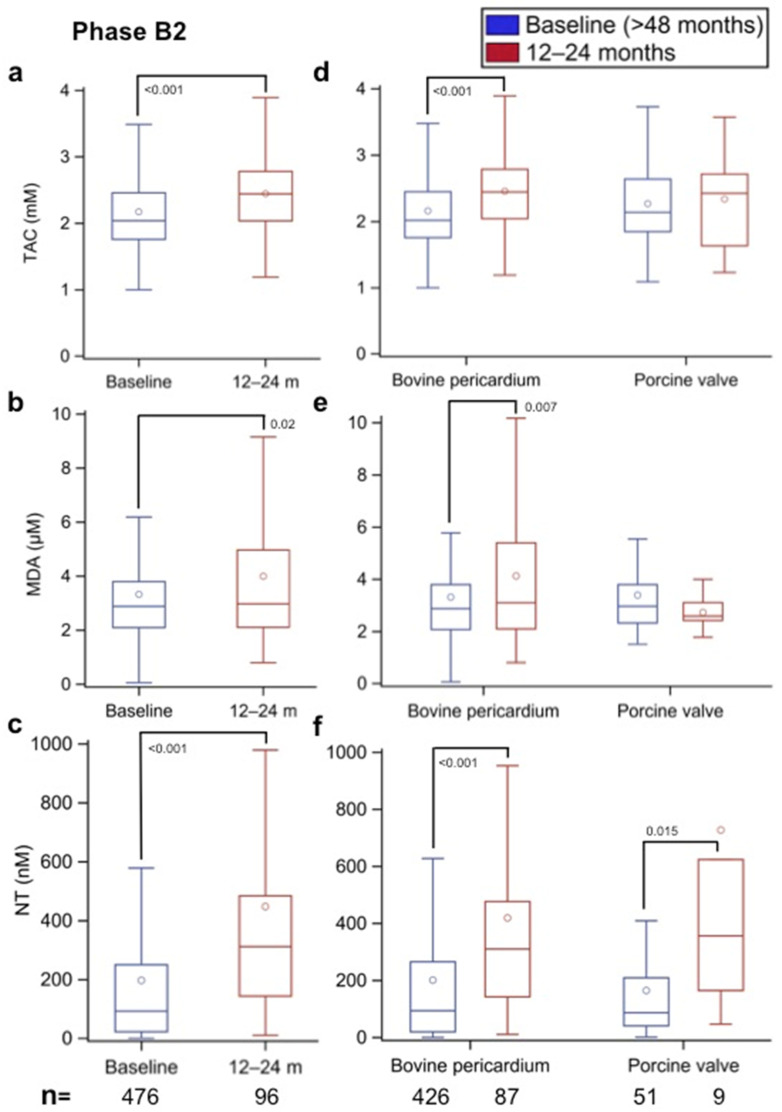
(**a**,**d**) Total antioxidant capacity (TAC, in mm), (**b**,**e**) malondialdehyde (MDA, in µm), and (**c**,**f**) nitrotyrosine (NT, in nm) values in Phase B2 patients with all types of implanted aortic biological heart valves (BHVs) and in Phase B2 patients with implanted bovine pericardium and porcine aortic BHVs, respectively, at the time of recruitment (>48 months after surgery) and after 12–24 months of follow-up (>60–72 months after aortic valve implantation). The numbers within the figures represent the *p*-values, and the numbers at the bottom of the figures represent the number of assessed patients. Multiple comparison adjustments were performed with Dunnett’s and Tukey–Kramer’s procedures.

**Figure 7 biomolecules-12-01606-f007:**
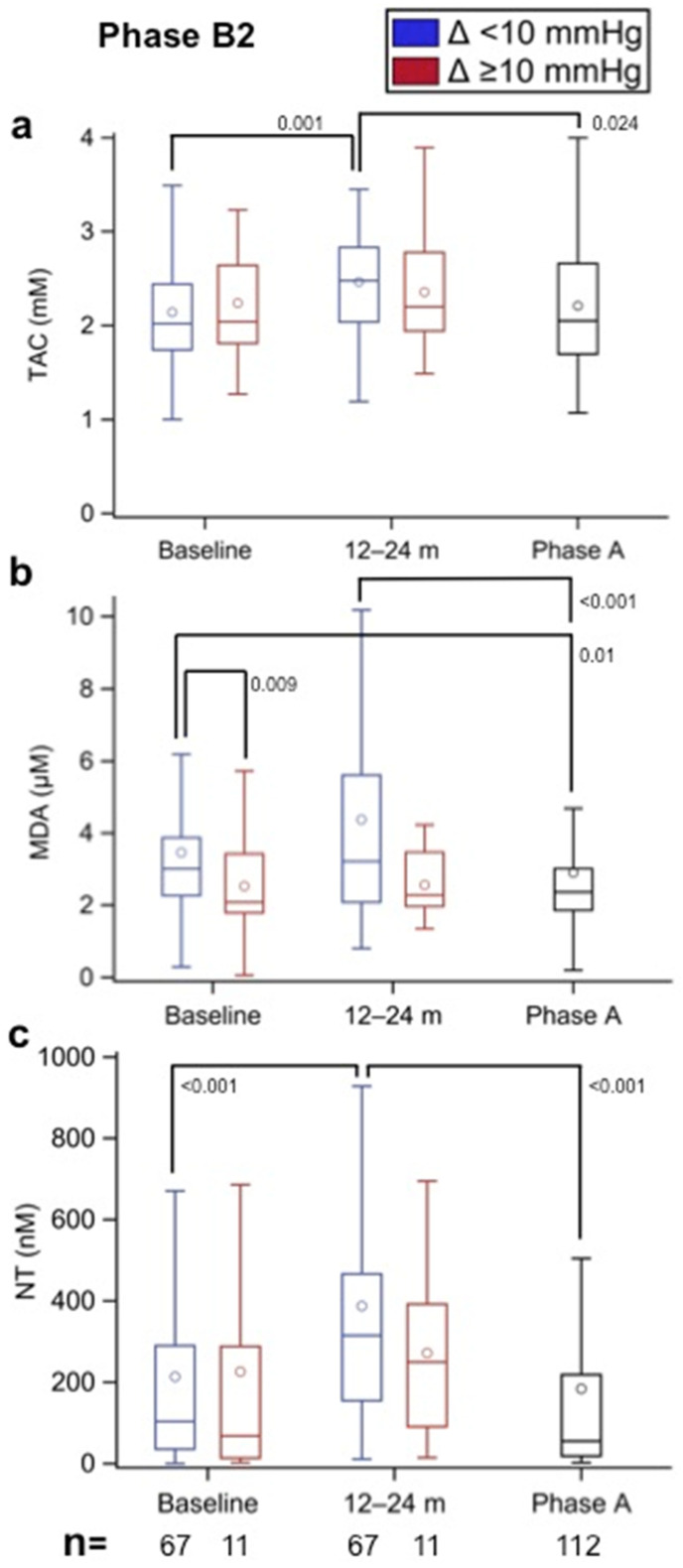
(**a**) Total antioxidant capacity (TAC, in mm), (**b**) malondialdehyde (MDA, in µm), and (**c**) nitrotyrosine (NT, in nm) values in Phase A patients (established SVD) and in Phase B2 patients with implanted aortic biological heart valves (BHV) according to the observed increment (Δ) in mean aortic prosthesis gradient (Δ < 10 mmHg—without incipient SVD; Δ ≥ 10 mmHg—with incipient SVD) between the baseline (>48 months after aortic valve implantation) and 12–24 months of follow-up (>60–72 months after aortic valve implantation). Therefore, each patient acted as his or her own control (e.g., paired results). The numbers within the figures represent the *p*-values, and the numbers at the bottom of the figures represent the number of assessed patients. Multiple comparison adjustments were performed with Dunnett’s and Tukey–Kramer’s procedures.

**Figure 8 biomolecules-12-01606-f008:**
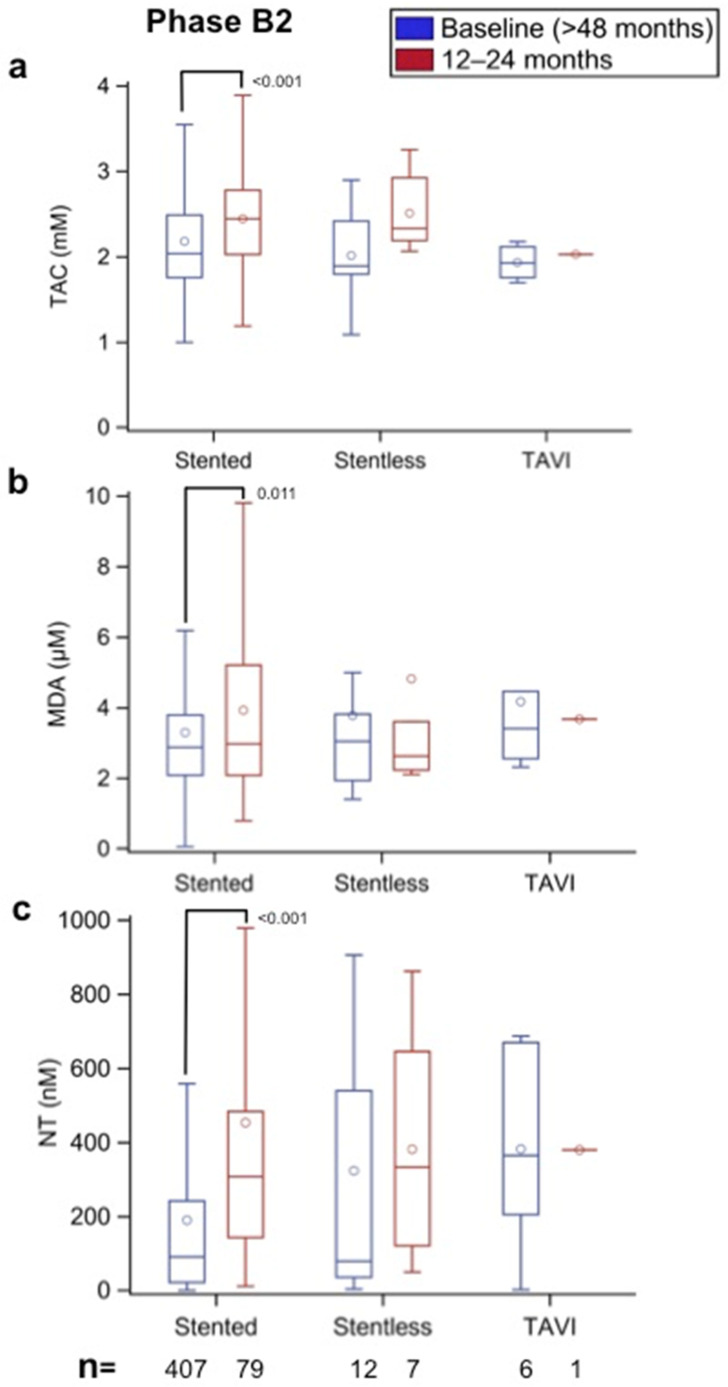
(**a**) Total antioxidant capacity (TAC, in mm), (**b**) malondialdehyde (MDA, in µm), and (**c**) nitrotyrosine (NT, in nm) values in Phase B2 patients with bovine pericardium aortic biological heart valves according to the type of framework (stented, stentless, TAVI) at the time of recruitment (>48 months after aortic valve replacement) and after 12- to 24-month follow-up (>60–72 months after aortic valve implantation). The numbers within the figures represent the *p*-values, and the numbers at the bottom of the figures represent the number of assessed patients. Multiple comparison adjustments were performed with Dunnett’s and Tukey–Kramer’s procedures.

**Table 1 biomolecules-12-01606-t001:** Patients’ characteristics at the time of entering the study and surgical data.

	Phase A	Phase B1			Phase B2		
		Control CABG	Control Mechanical Prostheses	Biological Prostheses	Control CABG	Control Mechanical Prostheses	Biological Prostheses
**PATIENTS** (number)	112	49	37	301	45	36	477
Male—total no. (%)	59 (52.3)	37 (75.5)	27 (77.1)	175 (58.5)	41 (91.1)	25 (71.4)	328 (68.8)
Age—yr, mean (SD)	76.9 (11.0)	69.4 (7.6)	59.6 (12.1)	74.8 (7.8)	76.2 (5.8)	70.7 (9.5)	79.1 (7.4)
Diabetes mellitus—total no. (%)	30 (26.8)	19 (30.6)	6 (17.1)	89 (29.8)	10 (22.2)	12 (34.3)	132 (27.9)
Hypertension—total no. (%)	90 (80.4)	41 (85.4)	24 (68.6)	251 (84.2)	35 (79.6)	23 (65.7)	400 (84.9)
MI—total no. (%)	NR	49 (100)	1 (2.7)	16 (5.4)	45 (100)	0 (0)	36 (7.6)
NO donors—total no. (%)	5 (4.5)	4 (8.2)	0 (0)	11 (3.7)	NR	NR	18 (3.8)
Statins—total no. (%)	64 (57.1)	33 (67.3)	15 (40.5)	174 (57.8)	NR	NR	284 (59.5)
OAC—total no. (%)	30 (26.8)	5 (10.2)	6 (16.2)	43 (14.3)	NR	NR	105 (22.0)
APT—total no. (%)	48 (42.9)	35 (71.4)	7 (18.9)	111 (36.9)	NR	NR	238 (49.9)
**SURGICAL DATA**							
Valve Surgery + CABG—total no. (%)	14 (12.5)	NR	NR	NR	NR	NR	NR
Cross-clamp time—min							
Median	NR	36.6	48.6	43.2	NR	NR	NR
IQR		33.0–44.4	38.4–57.6	33.0–57.0			
CPB time—min							
Median	NR	69.0	63.9	58.8	NR	NR	NR
IQR		60.6–75.6	52.2–72.6	44.4–72.0			

APT, antiplatelet drugs; CPB, cardiopulmonary bypass; IQR, interquartile range; MI, myocardial infarction; NO, nitric oxide; NR, not recorded; OAC, oral anticoagulant drugs.

**Table 2 biomolecules-12-01606-t002:** Types of biological prostheses implanted.

Type of Prostheses	Phase A (*n* = 112)	Phase B1 (*n* = 301)	Phase B2 (*n* = 477)
**Porcine Xenograft**	**16**	**7**	**51**
Mosaic (stented)	9	2	36
Epic & Epic Supra (stented)	2	0	11
Hancock II (stented)	1	5	0
Shelhigh (stentless)	2	0	0
Freestyle (stentless)	0	0	2
O’Brien (stentless)	1	0	2
**Bovine Pericardium**	**96**	**267**	**426**
Perimount Magna & Magna Ease (stented)	26	136	297
Mitroflow (stented)	67	32	111
Trifecta (stented)	0	25	0
Solo (stentless)	1	30	12
Perceval (sutureless)	0	22	0
Sapien (TAVI)	2	20	6
Jena (TAVI)	0	2	0
**Equine Pericardium**	**0**	**22**	**0**
3f (stentless)	0	15	0
Enable (sutureless)	0	7	0
**Porcine Pericardium**	**0**	**5**	**0**
CoreValve (TAVI)	0	4	0
Acurate neo/TF (TAVI)	0	1	0

## Data Availability

Not applicable.

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
