# Peer review of "Oxidative Stress in Structural Valve Deterioration: A Longitudinal Clinical Study"

_biomolecules, 2022, doi:10.3390/biom12111606_

Round 1

Reviewer 1 Report

The manuscript by Galinanes et al. analyses the concentration of MDA and NT, as well as the total antioxidative capacity in the serum of patients with bioprosthetic heart valves and subsequent structural valve deterioration.

The study is exceptionally well written. The introduction section is kept highly informative and appropriate. The material and methods section is described in detail, especially the used statistical tests. The results are adequately presented, really interesting and have a high novelty value. The discussion is also well formulated and all issues have been sufficiently discussed. The section on the limitations of the study is also particularly noteworthy.

Despite all this, a few minor questions arise for the reviewer:

·       The patient population includes some patients with diabetes. It is assumed that in diabetics there is increased oxidative stress in the vessels. Some anti-diabetic drugs are therefore also known to have an anti-oxidant effect, such as metformin. Can you tell which of the patients received which anti-diabetic medication and if so, could this have had an influence on the results? The same applies to statins

·       Minor Point: Uniform scales allow a better comparison of the measured concentrations between the groups.

·       Line 536: I am a little confused about the authors' statement that lipid oxidation and protein nitration cause oxidative stress. Isn't it rather the other way round, so that oxygen radicals produced within the valve lead to oxidative changes in lipids and proteins systemically? And these functionally impaired proteins and lipids lead to dysfunction?

Author Response

We thank the Reviewer for the interesting comments and important suggestions and for the positive rating of our manuscript. We have answered all the questions and made the suggested changes as described below:

(i) “The patient population includes some patients with diabetes. It is assumed that in diabetics there is increased oxidative stress in the vessels. Some anti-diabetic drugs are therefore also known to have an anti-oxidant effect, such as metformin. Can you tell which of the patients received which anti-diabetic medication and if so, could this have had an influence on the results? The same applies to statins”:

This is an important point raised by the Reviewer and, accordingly, we have re-analysed the data regarding the effect of the treatments on oxidative stress. The Supplementary Tables 4 and 5 show that the treatment with the antiglycemic drug metformin had a significant effect on the NT but not in the MDA serum levels. NT values were significantly lower in patients treated with metformin than in those without treatment before BHV implantation and during the first 6-month after implantation. However, beyond the 12-month follow-up the NT increased to a similar degree in both groups, a trend that was maintained for the remainder of the follow-up (Phase B2 patients). Interestingly, once SVD developed (Phase A) the NT was reduced again to values similar to ones seen at baseline with a greater reduction in the metformin-treated group, although the difference did not achieve statistical significance (p=0.079). Other medications did not show a noticeable effect on oxidative stress markers (data not shown). We have inserted this statement in lines 469-480 of the revised manuscript. We also have modified the title of the subheading 3.8. in line 462 as: “3.8. Effect of comorbidities and medical treatment on oxidative stress”. In addition, we have incorporated the following comment into the discussion (lines 519-525 of the revised manuscript): “A thesis further supported by the finding that the antiglycemic drug metformin reduces NT levels before BHV implantation and during the first 6 months of follow-up. Thereafter, the NT values continue to rise with disappearance of the beneficial effect of metformin, only to be reduced again when SVD develops. Indeed, the antioxidant effect of metformin has been previously described in endothelial cells [32] and in patients with type 2 diabetes [33].” Therefore, two new references [32 and 33] have been incorporated to the revised manuscript.

(ii) Uniform scales allow a better comparison of the measured concentrations between the groups”: As suggested by the Reviewer, the scales used for the oxidative stress biomarkers have been revised and changed where needed. Now in the revised version of the manuscript, the scales are uniformly used in all the Figures except in Figure 2b in which the Y axis remains at 15 µM due to the larger whisker on the MHV group at 24 months of follow-up.

(iii)  Line 536: I am a little confused about the authors' statement that lipid oxidation and protein nitration cause oxidative stress. Isn't it rather the other way round, so that oxygen radicals produced within the valve lead to oxidative changes in lipids and proteins systemically? And these functionally impaired proteins and lipids lead to dysfunction?”: The Reviewer is correct and we have removed the statement that appeared to be redundant and misleading (lines 562-563 of the revised manuscript). 

Reviewer 2 Report

Thank you for asking me to review this manuscript. This is an original article of oxidative stress marker detection in structural valve deterioration comparing patients without any deterioration after bioprosthetic valve implantation. The utility of oxidative stress markers as biomarker against bioprosthetic heart valve dysfunction is very useful , especially in early phase before significant heart failure symptom.Therefore, this manuscript could provide some important aspect about postoperative care in patients with bioprosthetic valve. However, there are few points to consider in this paper.

1. 
Table 1: the numbers of CABG-no/total no. (%) are missing. 

2. Table 2: what do you want to emphasize with bold? Please add the comment.

3. Methods: are the cohorts in Phase B1 or B2 same patients? If the same patients are controlled at the defined times, please comment about follow-up rate. If the cohorts are all different patients from several sample time, please explain this point.

4.Methods: were all CABG Patients as control group operated using CPB? If Off pump CABG are included, the comparison might be unreasonable. Please add the comment.

5. Line 265-273, 347,348, 404,407,409: Please unite the decimal places, maybe three is adequate. For example, “p<0.0001””p<0.001”  

Author Response

We thank the Reviewer for the positive rating of our manuscript and for the important suggestions and clarifications that we have incorporated to the revised version of the manuscript in the following manner:

(1)Table 1: the numbers of CABG-no/total no. (%) are missing”: The number of patients in the Control CABG group in Phases B1 and B2 is inserted in the first line of Table 1. The second line of Table 1 depicts the number of participant male patients and the percentage is shown into brackets. The number of study females is not shown in the Table because it can be directly obtained by subtraction from the male figures.

(2) Table 2: what do you want to emphasize with bold? Please add the comment.”: The bold was used to indicate the tissue origin of the different type of prosthesis used in the study.

(3) Methods: are the cohorts in Phase B1 or B2 same patients? If the same patients are controlled at the defined times, please comment about follow-up rate. If the cohorts are all different patients from several sample time, please explain this point.”: The participants in Phase B1 and B2 were different set of patients. According with the Reviewer’s recommendation, we have clarified the point in lines 147-153 as: “… and a Cohort B2, comprising a different set of patients with >48 months of BHV implantation to study the long-term oxidative stress response. This dual approach was required to cover the long period of time that in some cases may take SVD to develop, so that the occurrence of the phenomenon could be studied within the time frame of the 4.5 years that lasted the European project TRANSLINK that funded this clinical trial.“ Thank you to the Reviewer for the important clarification.

(4) Methods: were all CABG Patients as control group operated using CPB? If Off pump CABG are included, the comparison might be unreasonable. Please add the comment.”: All the patients in the CABG group were operated under cardiopulmonary bypass. Certainly, this is also a very important point that has been clarified in the revised version of the manuscript as follows (lines 153-154): “In both phases, patients with CABG surgery performed under cardiopulmonary bypass and MHV replacement served as controls.”

(5) Line 265-273, 347,348, 404,407,409: Please unite the decimal places, maybe three is adequate. For example, “p<0.0001”à”p<0.001” : We agree with the Reviewer’s suggestion of using a maximum of 3 decimal places for the p values and, therefore, the changes have been made where appropriate to the text and figures.